# Estimation of the Economic Threshold for the Fall Army Worm *Spodoptera frugiperda* (Lepidoptera: Noctuidae) in Short Stature Maize, Variety Delfín

**DOI:** 10.3390/insects16121219

**Published:** 2025-11-29

**Authors:** Néstor Bautista-Martínez, Omar Hernández-Romero, Everardo López-Bautista, Francisco Santos-González, Lauro Soto-Rojas, Job Alfonso López de Santana-Pimienta

**Affiliations:** 1Posgrado en Fitosanidad, Colegio de Postgraduados, Carretera México-Texcoco km 36.5, Montecillo, Texcoco 56230, Estado de México, Mexico; omar.hernandez@colpos.mx (O.H.-R.); rojo@colpos.mx (L.S.-R.); lopezdesantaana.job@colpos.mx (J.A.L.d.S.-P.); 2Facultad de Agricultura del Valle del Fuerte, Universidad Autónoma de Sinaloa, Ahome 81110, Sinaloa, Mexico; everardolob@uas.edu.mx; 3Departamento Investigación y Desarrollo, Bayer de México S.A. de C.V., Ciudad de México 11520, Mexico; francisco.santos1@bayer.com

**Keywords:** damage percentage, economic loss, FAW management

## Abstract

*Spodoptera frugiperda* is a key pest in Latin America, capable of causing great losses if it is not controlled on time. This study sought to estimate the economic threshold of the pest in short stature maize, Delfín variety, in Sinaloa, Mexico. We applied seven treatments with different levels of infestation and measured leaf damage, affected plants, and grain yield. Yield losses increased linearly, with 32.515 kg/ha less for each 1% of damage. Based on costs and the price of maize, we determined an economic threshold at 12.05% damaged plants. This is useful for improving management decisions, although we recommend validating it in other contexts.

## 1. Introduction

The fall army worm, *Spodoptera frugiperda* (J.E. Smith) (Lepidoptera: Noctuidae), is a migratory, polyphagous species with a high reproductive capacity, having more than 350 reported host plants. It produces several generations per year, and its development is favored by warm temperatures and continuous food availability. Females oviposit egg masses on the underside of leaves, and young larvae concentrate in the whorl of maize plants, where they cause the greatest damage [1]. In Mexico it causes the greatest losses in maize production, especially in tropical and neotropical regions of the country, where costs associated with its control can account for approximately 10% of total production costs [2].

The severity of damage caused by the fall army worm varies geographically. It has been reported that it is a more serious pest in coastal areas. In contrast, its importance decreases in areas at higher altitudes, above 2000 m [3].

In view of the threat posed by *S. frugiperda*, generation and validation of specific economic thresholds for maize in different agroecological conditions become prevailing necessities [1,4]. Application of economic thresholds allows growers to make informed decisions on when and how to intervene to control the pest, optimizing inputs (especially insecticides) and reducing production costs [4].

Determining precise economic thresholds for *S. frugiperda* in maize poses several challenges [4,5]. First, the relationship between pest density and yield can vary significantly depending on the plants’ growth stage and the larval instar [6]. For example, it has been observed that young maize plants can compensate for considerable early defoliation without significant reduction in yield [1]. However, damage at critical development stages, such as fringe (stigma) formation, can have a much greater impact [7].

Second, the manner in which pest pressure is reported (for example, number of larvae per plant, percent leaf damage) is often inconsistent among studies, making it difficult to compare results and generalize thresholds [4]. Third, the effectiveness and cost of the different control tactics (chemical insecticides, biopesticides, resistant varieties, cultural practices) have a direct influence on calculating the level of economic damage (LED) and, therefore, of the economic threshold (ET) [8]. Variations in input costs, maize market prices and the efficacy of phytosanitary products require continuous adaptation of economic thresholds to local conditions [8,9].

Despite these challenges, several studies have undertaken the task of estimating economic thresholds for *S. frugiperda* in maize in different regions of the world [8,9,10]. These studies have used different methodologies, from evaluating leaf damage with visual scales to conduct experiments of controlled infestation to quantify the relationship between pest density and yield losses [4,8].

A study conducted in Cuba, for example, proposed an alternative for estimating ET of *S. frugiperda* in maize based on data obtained over five years. The authors developed a regression model using a five-degree visual scale to determine leaf damage and calculate levels of economic damage [8]. The results suggest that LED could be reached with a relatively low percentage of plant loss, translating to an economic threshold based on the percentage of plants with 4–5-degree leaf damage [8].

Overton et al. (2021) published an overall review of the literature on yield losses caused by *S. frugiperda* in different crops and identified a wide range of loss values, highlighting the significant variability in the way levels of economic damage and action thresholds are reported. This variability underlines the need to conduct specific studies in each region for each production system to generate economic thresholds that are reliable and applicable locally [4]. Recently, we observed that, unlike other varieties or hybrids where the greatest risk of *S. frugiperda* infestation occurs around 45 days after planting, the Delfín variety maintains susceptibility up to the reproductive stage, just before fructification. This prolonged window of vulnerability suggests that short-stature hybrids may sustain larval feeding for a longer period, requiring specific economic threshold determinations adjusted to their phenology and growth habit.

For this reason, our objective was to estimate the economic threshold of *Spodoptera frugiperda* in Delfín variety maize in the state of Sinaloa, Mexico. We hypothesized that the economic threshold of *S. frugiperda* in the Delfín variety would be different from that reported for other maize genotypes.

## 2. Materials and Methods

### 2.1. Experiment Site

The study was conducted in the locality of Charay, municipality of El Fuerte, Sinaloa, Mexico, located at coordinates 26°00′57.3″ N and 108°50′23.9″ W. The soil at the site was classified as an alluvial Cambisol with a sandy–clay texture.

### 2.2. Plant Material

Maize (*Zea mays* L.) seeds of the hybrid variety Delfín were used as the main material for the study. Simultaneously, the DK-4050 variety was established as a regional control, since it is a hybrid widely used in the area. Both varieties were shown on the same day and under identical management practices (density, fertilization, and irrigation).

Evaluation began when both varieties reached the V3 stage (≈20 cm in height). Artificial infestation with neonate *S. frugiperda* larvae was applied exclusively to the Delfín plots, while DK-4050 remained without induced infestation to represent natural infestation conditions.

### 2.3. Biological Material

Neonate larvae of *Spodoptera frugiperda* (J.E. Smith) were obtained from the Biological Control Laboratory of the Colegio de Postgraduados, Campus Montecillo, from a colony maintained on an artificial diet under controlled conditions (27 ± 1 °C, 70 ± 5% RH, and a 12:12 h light–dark photoperiod).

Prior to hatching, egg masses were placed in Petri dishes (Ø = 10 cm) containing artificial diet to prevent cannibalism. Newly hatched larvae were transported in ventilated containers to the experimental site and kept at ambient temperature during infestation.

Each plant selected for infestation received two to three neonate larvae, deposited directly into the central whorl using fine-tipped entomological forceps. The larvae were released individually and randomly onto previously identified plants, avoiding concentration within a single row. Each infested plant was marked with plasticized ribbon for recognition in subsequent evaluations.

The DK-4050 variety was not artificially infested and was maintained under natural infestation conditions.

### 2.4. Experimental Design

Seven treatments were evaluated in a completely randomized design with three replications per treatment. Each experimental unit covered an area of 115 m^2^ (12 rows × 12 m, with 0.8 m between rows) and included 400 useful plants.

The six main treatments corresponded to increasing levels of induced infestation (0–25%) in the Delfín variety, defined by the number of infested plants within each 400-plant unit (Table 1). The infestation percentage was controlled through random selection of plants and precise application of the required number of larvae per plant (2–3), achieving equivalent larval densities ranging from 2.5 to 7.5 larvae/m^2^ depending on the treatment.

Treatment T1 (0–2%) remained without induced infestation, serving as an internal control to quantify natural infestation, whereas T7 (DK-4050, natural infestation) was included as a regional control to compare varietal response under natural *S. frugiperda* pressure.

### 2.5. Evaluation of the Experiment

Evaluations were conducted at 20, 40, and 60 days after infestation (dai). On each date, all 400 useful plants per plot were inspected, recording:− The percentage of damaged plants;− The severity of foliar damage determined using the scale [5].

Severity values were transformed into percentages using the Townsend and Heuberger formula, as follows:$$PS=\;\left(\frac{\sum_{v=0}^{9}\left(n_{v}\;(V)\right)}{\left(9)(400\right)}\right)\;(100)$$
where *PS* = percent severity; $n_{v}$ = number of plants with class value V; and V = numerical damage value according to the Davis scale.

### 2.6. Determination of Yield

Yield per treatment was obtained by weighing the grain harvested from the 400 useful plants in each experimental unit and extrapolating the values to a per-hectare basis. The average number of ears characteristic of the Delfín variety was considered, and for the regional control, the corresponding value of DK-4050 under its natural infestation condition.

### 2.7. Data Analysis

A one-way ANOVA was applied for the variable’s percentage of damaged plants, severity, and yield. When PR > F < 0.05, Tukey’s test (α = 0.05) was performed for mean comparison. Severity values calculated using the Townsend–Heuberger formula were based on the scale [5].

A linear regression model was fitted between the percentage of damaged plants and yield loss (kg ha^−1^) to estimate the relationship between damage and productivity. All analyses were performed in R version 4.0.2.

## 3. Results

### 3.1. Percentage of Damage in Plants

Table 2 presents the percentages of plants damaged by the fall army worm at three moments of crop development. Treatment T1 (without induced infestation) maintained the lowest levels of damage in all the evaluations with values of 0.00%, 1.58% and 2.58%, respectively. Treatment T6 (25% infestation) showed the highest percentages of damaged plants, with 24.25%, 24.58% and 24.33% at each evaluation, placing it consistently in the highest statistical group (“a”). The intermediate treatments (T2 to T5) had progressive increments in damage according to the established level of infestation with significant statistical differences among them, while treatment T7 (variety DK-4050 with natural infestation) had initial behavior similar to T2, with 4.25% in the first evaluation but reached 11.42% in the third evaluation, placing it in the same statistical group as T3 (10% artificial infestation). The analysis of variance revealed significant differences among treatments for the percentage of damaged plants at 20, 40, and 60 days after infestation (F~6, 14~ = 1180, 500.2, and 339.1, respectively; *p* < 0.0001 for all), confirming a strong and consistent relationship between infestation level and leaf damage.

### 3.2. Severity in Plants

Table 3 shows the values of severity caused by *S. frugiperda*, based on the scale [5]. Treatment T1 (without induced infestation) had the lowest levels of severity with values of 0.00%, 0.88% and 1.85% in the three evaluations, respectively. Treatment T6 (25% infestation) had the highest values with 7.15%, 14.68% and 17.99%, corresponding to the statistical group with greatest severity (“a”) in all the evaluations. The intermediate treatments (T2 to T5) exhibited gradual increments in severity as the level of infestation increased, maintaining them in clearly, statistically different groups. Moreover, the treatment with variety DK-4050 infested naturally began with a severity of 0.71% and reached 7.82% in the third evaluation, placing it in the same group as treatment T3 (10% artificial infestation). The analysis of variance revealed significant differences in severity among treatments at 20, 40, and 60 days after infestation (F~6, 14~ = 585.9, 1292, and 1200, respectively; *p* < 0.0001 for all). This indicates that the severity of foliar damage increased proportionally with the level of infestation.

### 3.3. Determination of Yield

As can be seen in Table 4, mean yield per hectare decreased as the percentage of infestation by the fall army worm increased. Treatment T1, with a minimal level of infestation (2.58%) had the highest yield, with 12,199.18 kg/ha. From this point on, the treatments with induced infestations showed progressive reductions in yield. Treatment T2 (4.92%) produced 12,064.07 kg/ha, 135.11 kg less than T1. In T3 (9.58%), yield was 11,917.92 kg/ha, a reduction of 281.26 kg, relative to T1, while T4 (14.75%) had a yield of 11,766.25 kg/ha, 432.93 kg less than T1. In T5 (19.33%) yield decreased to 11,578.74 kg/ha, an accumulated loss of 620.44 kg. Finally, treatment T6 (24.33%) had the lowest yield, with 11,316.78 kg/ha, a reduction of 882.40 kg. Treatment T7 (natural infestation), with an average infestation of 11.42%, produced a yield of 11,912.40 kg/ha, like treatment T3, indicating that natural infestation reached levels comparable to artificial infestation of 10%.

A strong linear relationship was found between the percentage of damaged plants and yield loss (y = 32.515x; R^2^ = 0.9874). This indicates that for every 1% increase in damage, yield decreases by approximately 32.5 kg ha^−1^. The high R^2^ confirms the robustness of the regression model, explaining 98.7% of yield variability, and supports its predictive potential for estimating losses under similar agroecological conditions (Figure 1).

### 3.4. Economic Threshold

Analysis of the relationship between leaf damage and yield led to identification of the starting point from which it is economically viable to apply control measures against *Spodoptera frugiperda*. Based on a reference maize price of US $325 t^−1^ and an average control cost of US $127 ha^−1^ (equivalent to two insecticide applications), the yield loss matching that cost was estimated at 391.6 kg ha^−1^.

These values were obtained from regional averages reported by producers and field technicians in northern Sinaloa during the 2024 agricultural cycle, reflecting the market prices of conventional maize and the local costs of chemical control using products widely employed in the area. The cost of two insecticide applications—including agrochemical, labor, and machinery use—is a common practice in the region. Field observations indicate that the pest may persist until stages close to fructification in the Delfín variety; however, under a timely management scheme, two well-scheduled applications are generally sufficient to maintain *S. frugiperda* populations below the economic threshold.

Substituting this value into the regression equation (y = 32.515x) yields X = 12.05%. This percentage represents the level of economic damage (LED), the point at which yield loss equals the cost of control. The economic threshold (ET) or “action level” should therefore be established below 12.05%, enabling timely interventions before reaching the LED and preventing irreversible economic losses.

From this point, the application of phytosanitary measures against the pest is justified to avoid greater losses, although this will depend on future fluctuations in management costs and market prices.

## 4. Discussion

The results of this study confirm a strong linear relationship between the percentage of plants damaged by *Spodoptera frugiperda* and yield loss in maize (Delfín variety) (R^2^ = 0.9874), demonstrating that this variable can be used as a reliable indicator of the pest’s economic impact under controlled conditions. However, the results should be interpreted considering the limitations of the experiment, which was conducted in a single location and agricultural cycle, without incorporating natural variations in larval density, natural enemies, or climatic factors that may alter the population dynamics of the fall armyworm and the crop’s recovery capacity.

Despite these limitations, the study provides a solid basis for estimating variety-specific economic thresholds, an essential condition for optimizing integrated pest management. The value obtained (12.05% of damaged plants) offers a practical reference for reducing unnecessary insecticide applications and promoting a more rational use of phytosanitary inputs. Future studies should validate this threshold under different environments, planting seasons, and genotypes, and integrate ecological and economic factors to develop dynamic decision-making models applicable to various maize-producing regions in Mexico.

In this context, the results obtained are consistent with previous reports on losses caused by *S. frugiperda* under different regions and management conditions, allowing comparisons of their relative magnitude and validation of the trend observed in this study.

These findings align with multiple studies. For example, ref. [10] reported losses of 28.2% in yield in plots infested with six larvae per plant, 7.2% higher than our results under moderate infestation of 24.33% but conditions of greater biological pressure. Likewise, ref. [12] reported losses of 45% with 100% infestation, and [13] observed reductions of 22.6% in plants infested in the first two weeks of vegetative development, highlighting that early denser infestations generate greater losses.

Although the results we obtained in this study coincide with the literature that *S. frugiperda* directly reduces maize yield, genotype plays an important role in yield losses since certain cultivars possess greater natural tolerance. Such is the case in Mozambique, where losses that varied between 17.7% and 55.6% were observed, depending on the genotype evaluated; PAN53 and AZ523 were those of lowest and highest yield, respectively, under natural infestation [14]. Moreover, the phenological stage in which infestation occurs also has a significant influence. In South Africa losses of up to 56.8% have been reported when attack occurs in early stages of development, while later infestations caused lower losses (26.5%) [15]. Also, in Pakistan a positive correlation was observed between high temperatures and larger larval population, with densities of up to 15.41 larvae per plant and losses of more than 9 thousand kg/ha in some varieties [16]. Type and time of agronomic management are also determinant. In India, it was found that control strategies can reduce losses from 43.6% to only 6%; this is dependent on the effectiveness of the treatment and time of application [17].

In terms of severity, the highest values found in this experiment (17.99%) are comparable with the 14.68% reported for the same treatment (T6) in the second evaluation (40 dai), with leaf damage scores in the order of 17.5–21.5% reported by [18], who documented a loss of 27.06% in yield and reductions of up to 38.87% in number of grains per ear.

In yield, we observed an average linear loss of 32.515 kg/ha with every 1% increase in the percentage of damaged plants (R^2^ = 0.9874). This model coincides notably with that proposed by [12], who used the formula Y = 87.84 − 0.384 PI (PI = % infested plants), which implies a loss of 384 kg/ha with each 10%, a difference of only 59 kg/ha. This shows the solidity of the local predictive model.

Regarding the economic threshold for intervention, we established it at 12.05% damaged plants. This value was obtained by solving the regression equation starting from the economic loss equivalent to the cost of two applications of insecticides ($2350/ha). This threshold is comparable with the 11% to 12% reported by Overton et al. (2021) in Colombia and the lower pest population densities of 3.32 to 4.44 larvae per plant in the V4 stage reported by [10], which resulted in losses in yield of up to 28% [4].

This threshold is also found below the upper range reported by [19], who calculated that the lowest population density of the species that should be present in the crop is between 23% and 63% infested plants, in function of higher control costs and crops with lower expected yield.

Variety DK-4050 (T7), in conditions of natural infestation (11.42%), had a yield of 11,912.40 kg/ha, almost identical to treatment T3 (10% artificial infestation), suggesting that natural pressure was sufficient to cause damage comparable to that of controlled experimental management. This coincides with the findings of [14], who found losses of up to 55.6% in conditions of natural infestation, depending on the genotype used.

## 5. Conclusions

This study estimated, for the first time, the economic threshold of *Spodoptera frugiperda* in short stature maize, variety Delfín under the agroecological conditions of northern Sinaloa, Mexico. We observed a direct, significant, highly consistent relationship between the percentage of damaged plants and yield loss, with an estimated loss of 32.515 kg/ha for every 1% increment in leaf damage.

With this relationship and considering current grain prices and costs of control, we determined an economic threshold of 12.05% damaged plants, beyond which implementing control measures against *S. frugiperda* becomes economically justified. This value coincides with thresholds reported in other international studies, reinforcing its practical applicability. Future research should validate this threshold across different localities, planting cycles, and varying biotic pressures to ensure consistent performance under changing production and market conditions.

Our results contribute key information for decision-making in integrated pest management programs (IPM), promoting a more rational and efficient use of phytosanitary inputs. We recommend validating this threshold in different growing cycles, localities and conditions of biotic pressure, as well as evaluating its adjustment in function of fluctuations in prices and production costs.

## Figures and Tables

**Figure 1 insects-16-01219-f001:**
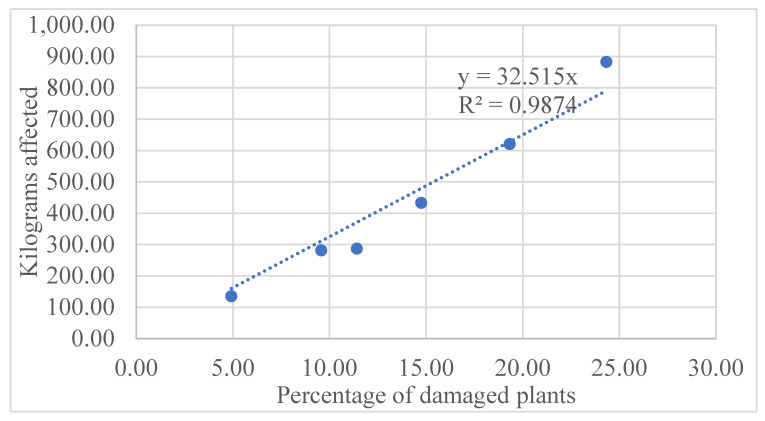
Linear relationship between the percentage of maize plants damaged by *S. frugiperda* and yield loss (kg ha^−1^) in the Delfín variety.

**Table 1 insects-16-01219-t001:** Treatments evaluated to determine the economic threshold for *S. frugiperda* in maize, variety Delfín.

Treatments	Percentage Infestation	Number of Infested Plants of 400 Plants
T1.	0–2%	0 plants
T2.	05%	20 plants
T3.	10%	40 plants
T4.	15%	60 plants
T5.	20%	80 plants
T6.	25%	100 plants
T7.	DK-4050	Natural infestation

**Table 2 insects-16-01219-t002:** Percentage of maize plants damaged by *S. frugiperda* at three developmental stages of the crop (Delfín variety).

Treatment	20 dai (%) (Pr > F < 0.0001)		40 dai (%) (Pr > F < 0.0001)		60 dai (%) (Pr > F < 0.0001)	
T1	0	f	1.58	g	2.58	f
T2	4.67	e	4.83	f	4.92	e
T3	9.67	d	9.58	d	9.58	d
T4	14.42	c	14.58	c	14.75	c
T5	19.25	b	19.33	b	19.33	b
T6	24.25	a	24.58	a	24.33	a
T7	4.25	e	7.42	e	11.42	d

Note: Means within the same column followed by different letters are statistically different (Tukey’s test, α = 0.05). dai = days after infestation.

**Table 3 insects-16-01219-t003:** Percentage of severity (PS) of foliar damage, calculated using the Townsend and Heuberger (1943) [11] formula based on the scale [5], in maize plants (Delfín variety) infested with *S. frugiperda*.

Treatment	20 dai (%) (Pr > F < 0.0001)		40 dai (%) (Pr > F < 0.0001)		60 dai (%) (Pr > F < 0.0001)	
T1	0	f	0.88	g	1.85	f
T2	1.11	e	2.98	f	3.81	e
T3	2.5	d	5.94	d	7.41	d
T4	3.96	c	8.6	c	10.62	c
T5	5.39	b	12.3	b	14.89	b
T6	7.15	a	14.68	a	17.99	a
T7	0.71	e	4.41	e	7.82	d

Note: Means within the same column followed by different letters are statistically different (Tukey’s test, α = 0.05). dai = days after infestation.

**Table 4 insects-16-01219-t004:** Mean yield of treatments at different percentages of maize plants damaged by *S. frugiperda* in the Delfín variety.

Treatment	Percentage of Damaged Plants	Average Yield/ha	Reduction in Yield (kg)
T1	2.58	12,199.18	
T2	4.92	12,064.07	135.11
T3	9.58	11,917.92	281.26
T4	14.75	11,766.25	432.93
T5	19.33	11,578.74	620.44
T6	24.33	11,316.78	882.40
T7	11.42	11,912.40	286.78

## Data Availability

The original contributions presented in this study are included in the article. Further inquiries can be directed to the corresponding author.

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
