# Peer review of "Estimation of the Economic Threshold for the Fall Army Worm Spodoptera frugiperda (Lepidoptera: Noctuidae) in Short Stature Maize, Variety Delfín"

_insects, 2025, doi:10.3390/insects16121219_

Round 1
Reviewer 1 Report
Comments and Suggestions for Authors
This study focuses on determining the economic threshold (ET) of Spodoptera frugiperda (FAW) on the short stature maize variety Delfín in Mexico. The experimental design follows a conventional approach, and the study identified an economic threshold of 12.05% damaged plants, which provides some theoretical and practical value for the management FAW.
However, the manuscript suffers from unclear presentation, making it challenging for readers to follow the logic and interpret key findings.
- Regarding the maize variety Delfínstudied in the paper, the authors describe it as a special Short Stature Maize, yet fail to provide basic information about this variety, such as its planting status in Mexico (e.g., whether it is a major maize variety locally). Such details are critical to supporting the study’s rationale. Additionally, the choice of DK405 as the control variety requires clarification regarding its key characteristics, including whether it is a short stature variety.
- The Methods section lacks critical details, significantly hindering comprehension of the study. Key clarifications are needed:Were DK4050 and Delfín planted simultaneously? During the FAW infestation process, was DK4050 also at the V3 phenological stage? Since T1 (non-inoculated) may still be naturally infested with FAW, What is the rationale for including the DK4050 treatment?
- The Methods section lacks critical details on FAWinfestation procedures, which hinders understanding of the experimental design. Specific clarifications are required: (1)What methods were used to control the percentage of damaged plants within each experimental plot during the inoculation? (2) Pest population density is a key factor (complementary to the percentage of damaged plants), yet it was not reported in Section 2.4 ("Experimental Design").(4) How were the FAW released. Were they placed randomly or clustered together? (5) For the three damage assessments: Were all 400 plants examined, or was a sampling method used? These omissions obscure the reproducibility of the experiment and limit the reliability of the results.
- Please verify the correctness of the formula’s notation (lines L130–132) and confirm the accuracy of variable definitions. Theparameter "v" have two distinct meanings in line 131-132. The first definition of "v" intended to be equivalent to that of "nv". This is a clear error.
- There is an inconsistency between Table 2’s title and its described content in the text. The table title is "Leaf damage to plants," but Line 52 states, "Table 2 presents the percentages of plants damaged by FAWat three stages of crop development."
- Does "Severity of damage" in Table 3 refer to the "PS"?
- Several issues require attention in Table 4 and Figure 1:(1) in Table 4 Columns 2 and 3 lack units. Additionally, "tratamiento" in Column 1 is a clear spelling error (should be "Treatment").(2) The term "average infestation" is ambiguous. Based on Figure 1, it likely refers to the "percentage of damaged plants" (as implied by the data pattern). We recommend confirming the precise definition of "average infestation" in the text to avoid misinterpretation.(3)The title of Figure 1 specifies that the data pertain to the Delfín variety, but the data presented in the figure include both the Delfín variety and the T7 control. (4)The quality of Figure 1 is poor.
- When calculating ET, the authors considered maize price and control cost (6,000/ton for maize and 3,250/ha for control)—it would be helpful to clarify how these specific values were obtained, including whether they are representative of conventional maize varieties or specifically the Delfín short-stature variety. Additionally, it should be clarified whether different maize varieties (e.g., Delfín vs. conventional) exhibit price disparities and whether control costs are uniform across varieties. Given that “the Delfín variety has the highest risk of infestation continuing up to just before fructification”, does this also imply that only two control applications are practically needed in production?.
- The Discussion section lacks depth, relying excessively on comparisons with previous studies; it should more critically highlight the limitations of the present study and propose future research directions.
Author Response
This study focuses on determining the economic threshold (ET) for Spodoptera frugiperda (FAW) in the dwarf maize variety Delfín in Mexico. The experimental design follows a conventional approach, and the study identified an economic threshold of 12.05% of damaged plants, which provides theoretical and practical value for FAW management.
However, the manuscript suffers from an unclear presentation, making it difficult for readers to follow the logic and interpret the key findings.
1. Regarding the Delfín maize variety studied in the article, the authors describe it as a special, short-stature maize, but they do not provide basic information about this variety, such as its planting status in Mexico (for example, whether it is an important maize variety locally). These details are crucial to support the study's justification. Furthermore, the choice of DK405 as a control variety requires clarification regarding its key characteristics, including whether it is a short-stature variety. Key information describing the variety was added, explaining why the Delfín variety was used, given that it is a new market introduction. On the other hand, the DK405 variety is one of the most frequently used locally; therefore, comparing a new biological material with one that has proven its adaptation to the region is considered the most appropriate approach.
2. The Methods section lacks critical details, which considerably hinders the understanding of the study. Key clarifications are required: Were DK4050 and Delfín sown simultaneously? During the FAW infestation process, was DK4050 also at the V3 phenological stage? Given that T1 (not inoculated) may still be naturally infested with FAW, what is the justification for including the DK4050 treatment? The document adds that both varieties were sown simultaneously, that the infestation was carried out simultaneously and randomly, and that the DK4050 variety was naturally infested.
3. The Methods section lacks crucial details regarding the fall armyworm infestation procedures, hindering the understanding of the experimental design. Specific clarifications are required: (1) What methods were used to control the percentage of damaged plants in each experimental plot during inoculation? (2) Pest population density is a key factor (complementary to the percentage of damaged plants), but it was not reported in Section 2.4 ("Experimental Design"). (4) How were the fall armyworms released? Were they placed randomly or grouped? (5) For the three damage assessments: Were all 400 plants examined, or was a sampling method used? These omissions hinder the reproducibility of the experiment and limit the reliability of the results. The methodology for infesting the plants with fall armyworm larvae has been added. Pest population density has been clarified in the experimental design section, and the range of densities according to the treatment has been discussed.
4. 31-1 Verify the correctness of the formula notation (lines L130-132) and confirm the accuracy of the variable definitions. The parameter "v" has two distinct meanings in line 132. The first definition of "v" was intended to be equivalent to that of "n v ". This is an obvious error. The comment regarding the parameter "v" was addressed, and the erroneous text was removed. In addition, the method of evaluating the experiment was clarified .
5. There is an inconsistency between the title of Table 2 and its content as described in the text. The table title is "Foliar Damage in Plants," but line 52 states: "Table 2 presents the percentages of plants damaged by FAW at three stages of crop development." The inconsistency in the table title has been corrected and adjusted to match the description.
6. Does the “Damage Severity” in Table 3 refer to “PS”? The same was done for this Table (3), where it is clarified that it is the percentage of severity of foliar damage.
7. Several aspects of Table 4 and Figure 1 should be considered: (1) In Table 4, columns 2 and 3 lack units. Furthermore, the word "tratamiento" in column 1 is a clear misspelling (it should be " Treatment "). (2) The term "average infestation" is ambiguous. According to Figure 1, it likely refers to the "percentage of damaged plants" (as can be inferred from the data pattern). We recommend confirming the precise definition of "average infestation" in the text to avoid misinterpretations. (3) The title of Figure 1 specifies that the data correspond to the Delfín variety, but the data presented in the figure include both the Delfín variety and the control T7. (4) The quality of Figure 1 is poor. Units from Table 4 have been added, and the text has been adjusted accordingly. The column name "percentage of plants" was changed to the suggested name "percentage of damaged plants" as it better represents the data shown in the column. The text clarifies that the data in Figure 1 represents the linear relationship between the percentage of damaged plants in the Delfin variety .
8. In calculating the ET (Efficient Entity), the authors considered the price of corn and the cost of control (6000/ton for corn and 3250/ha for control). It would be helpful to clarify how these specific values were obtained, including whether they are representative of conventional corn varieties or specifically of the short-stature Delfín variety. Furthermore, it should be clarified whether different corn varieties (e.g., Delfín vs. conventional) exhibit price disparities and whether control costs are uniform among them. Given that "the Delfín variety presents the greatest risk of infestation continuing until just before fruiting," does this also imply that only two control applications are needed in practical production? The origin of the corn price is clarified, relating it to the average price of national production (SIAP) and the prices reported by producers and technicians in the region. Additionally, the cost of control per hectare is calculated based on the value of inputs and labor in the producing region. This has nothing to do with the specific management of any variety, but rather with the overall crop management. Applications were made according to monitoring guidelines and in a timely manner, and it is noted that the application was carried out prior to fruiting because the fall armyworm is present until this phenological stage.
9. The Discussion section lacks depth and relies excessively on comparisons with previous studies; it should more critically highlight the limitations of the present study and propose future research directions. The first paragraph was rewritten into two to encompass the discussion of the interpretation of the results, including the study's limitations and factors that were not taken into account. Furthermore, it explains how the 12.05% value provides a reference point for management, based on the findings of this study.
Reviewer 2 Report
Comments and Suggestions for Authors
These are my main comments on the manuscript (Insects-3904550) entitled “Estimation of the Economic Threshold for the Fall Army Worm Spodoptera frugiperda (Lepidoptera: Noctuidae) in Short stature Maize, Variety Delfín”. This work investigates the economic threshold of S. frugiperda in short-stature maize. Following substantial revisions should be incorporated in the manuscript prior to acceptance.
1. I have concerns about the manuscript sections that I believe need to be addressed in order to improve its clarity.
2. Manuscript should be proofread by a native English speaker to enhance clarity and readability (i.e., Table 4).
3. In results, information about statistical methods is missing (F-value, degrees of freedom, and p-value).
4. I am concerned about the poorly elements in the discussion; authors must be provided a good debate for results interpretation.
5. A few points:
Ls.19-21: Revise this sentence to eliminate rewordiness.
Ls.40-44: More information about biological and ecological aspects of S. frugiperda is needed.
L.78: Overton et al. 2021 is the same reference 4?
Ls. 83-85: Any references?
Ls.92-93: Revise this sentence to eliminate rewordiness.
Ls.109-111: Summarize this sentence.
L.134: Delete “To determine yield per treatment,”
L.139: Delete “To analyze the data obtained in the study,”
Ls.143-144: This information is repetitive (see lines 128-132)
L.162: The statistical analysis revealed highly significant differences…And if it reveals lower differences, can they also be significant? Rewrite.
Ls.163, 178: By each Anova, provide statistical values (F-value, degrees of freedom, and p-value)
Table 4: Information that is not being analyzed statistically.
Ls.242:…the literature that…Which literature?
Manuscript should be proofread by a native English speaker to enhance clarity and readability.
Author Response
These are my main comments on the manuscript (Insects-3904550) entitled “Estimation of the economic threshold for the fall armyworm Spodoptera This study investigates the economic threshold for S. frugiperda ( Lepidoptera : Noctuidae ) in dwarf maize, variety Delfín. The following substantial revisions must be incorporated into the manuscript before acceptance.
- I have concerns about sections of the manuscript that I believe should be addressed to improve its clarity. The methodology sections were addressed to clarify how the experiment was conducted and to ensure its replicability. Additionally, the legends for tables and figures, which were being misinterpreted due to poorly worded titles, were clarified.
2. The manuscript should be reviewed by a native English speaker to improve clarity and readability (i.e., Table 4).
3. The results lack information on the statistical methods (F-value, degrees of freedom, and p- value). Information is available that is described in the paragraphs regarding the F-value within the text.
4. I am concerned about the deficient elements in the discussion; the authors should be provided with a good debate for the interpretation of the results. The first paragraph was reformulated into two, attempting to encompass the comments on the interpretation of the results found, where the limitations of the study are discussed as factors that were not taken into account. On the other hand, it is explained how the value of 12.05% offers a reference value for management, due to the results found in this study.
5. Some points:
Lines 19-21: Review this sentence to remove the reformulation. The sentence was reformulated for greater clarity.
Lines 40-44: More information is needed on the biological and ecological aspects of S. frugiperda. Biological information about the insect is added, such as that it is a migratory, polyphagous species, the number of hosts, and information on the generations per year and the favorable climatic conditions for it.
L.78: Is Overton et al. 2021 the same as reference 4? That reference is indeed from Overton, but it is stated that this author identified the loss values.
Ls . 83-85: Any references? Reference #2 is added as a source that justifies the information.
Ls.92-93: Review this sentence to remove the reformulation. The sentence is reviewed, corrected, and clarified.
Ls.109-111: Summarize this sentence. The sentence was summarized for clarity .
L.134: Remove “To determine the yield per treatment.” It was removed .
L.139: Remove “To analyze the data obtained in the study.” The sentence was reformulated, and the unnecessary text was removed.
Ls.143-144: This information is repetitive (see lines 128-132). The sentence was reformulated to remove the repetition.
L.162: The statistical analysis revealed highly significant differences… And if it reveals minor differences, can they also be significant? Rewrite. These sentences are rewritten, leaving only that they are significant.
Ls.163, 178: For each ANOVA , provide statistical values (F-value, degrees of freedom, and p- value).
Table 4: Information that is not being statistically analyzed.
Ls.242:… the literature that… What literature? It is clarified that the information refers to the cited literature, and the information is simplified.
Round 2
Reviewer 1 Report
Comments and Suggestions for Authors
The authors have addressed most of my comments, and I recommend acceptance in its current form.
Author Response
The changes recommended by reviewer 1 have been implemented.
Reviewer 2 Report
Comments and Suggestions for Authors
The authors have incorporated all suggestions and comments into the revised version; now the manuscript seems much clearer. There are some minor points to be corrected:
Ls.169-220: provide statistical values (F value, degrees of freedom, and P-value) to confirm the significant effect.
Author Response
The statistical values (F-value, degrees of freedom, and p-value) have been provided to confirm the significant effect.